# Chloroplast Localized FIBRILLIN11 Is Involved in the Osmotic Stress Response during Arabidopsis Seed Germination

**DOI:** 10.3390/biology10050368

**Published:** 2021-04-25

**Authors:** Yu-Ri Choi, Inyoung Kim, Manu Kumar, Jaekyung Shim, Hyun-Uk Kim

**Affiliations:** 1Department of Molecular Biology, Graduate School, Sejong University, Seoul 05006, Korea; choshinsungs@sju.ac.kr (Y.-R.C.); kiy88410@sju.ac.kr (I.K.); 2Department of Life Science, Dongguk University-Seoul, Ilsandong-gu, Goyang-si 10326, Korea; manukumar@dongguk.edu; 3Department of Bioindustry and Bioresource Engineering, Plant Engineering Research Institute, Sejong University, Seoul 05006, Korea; jkshim@sejong.ac.kr

**Keywords:** FIBRILLIN, kinase, chloroplast, osmotic stress, seed germination, ABA signaling pathway

## Abstract

**Simple Summary:**

The FIBRILLIN11 (FBN11) of Arabidopsis has a lipid-binding FBN domain and a kinase domain. FBN11 is present in chloroplasts and is involved in salt and osmotic stress responses during seed germination. In mannitol, the seed germination rate of the *fbn11* mutants significantly reduced compared to that of the wild type. The ABA-dependent and -independent stress response regulating genes were differentially expressed in *fbn11* mutants and wild-type when grown in mannitol supplemented medium. These results suggest that chloroplast localized FBN11 is involved in mediating osmotic stress tolerance through the signaling pathway that regulates the stress response in the nucleus.

**Abstract:**

Plants live in ever-changing environments, facing adverse environmental conditions including pathogen infection, herbivore attack, drought, high temperature, low temperature, nutrient deficiency, toxic metal soil contamination, high salt, and osmotic imbalance that inhibit overall plant growth and development. Plants have evolved mechanisms to cope with these stresses. In this study, we found that the *FIBRILLIN11* (*FBN11*) gene in Arabidopsis, which has a lipid-binding FBN domain and a kinase domain, is involved in the plant’s response to abiotic stressors, including salt and osmotic stresses. FBN11 protein localizes to the chloroplast. *FBN11* gene expression significantly changed when plants were exposed to the abiotic stress response mediators such as abscisic acid (ABA), sodium chloride (NaCl), and mannitol. The seed germination rates of *fbn11* homozygous mutants in different concentrations of mannitol and NaCl were significantly reduced compared to wild type. ABA-dependent and -independent stress response regulatory genes were differentially expressed in the *fbn11* mutant compared with wild type when grown in mannitol medium. These results suggest a clear role for chloroplast-localized FBN11 in mediating osmotic stress tolerance via the stress response regulatory signaling pathway in the nucleus.

## 1. Introduction

FIBRILLINs (FBNs) are structural proteins found in plastoglobules (PGs), which are lipoprotein particles present in chloroplasts [1]. They combine with carotenoids to form fibril structures within pepper chromoplasts [2]. They are named FIBRILLINs for their participation in microfibrils. They are also a major PG component in chloroplasts and constitute a large group of proteins known to play conserved roles in photosynthesis in cyanobacteria to higher plants [3,4]. Recent research suggests that FBNs are involved in the stress response and in the formation of plastid structures in plants [5,6,7,8,9,10,11,12,13,14]. These responses include disease resistance mechanisms, hormonal signaling, growth and development, and lipid transport between PGs and thylakoid membranes [11]. FBN localization within plastids is dependent on its isoelectric point [6] and hydrophobicity. According to previous studies, FBN1a, 1b, 2, 4, 7a, 7b, and 8 are in the PG core, FBN10 is in the PG-thylakoid, FBN5 is in the stroma, and FBN3a, 3b, 6, 9, and 11 are expected to be located in the thylakoid [15].

The role of FBN in abiotic stress has been studied for decades, but little is known about its function(s). *FBN* gene expression is regulated by diverse, complex mechanisms responsive to numerous stresses, including high temperature, low temperature, high light, drought, wounds, and herbicides [11]. In drought or cold stress, *FBN1a* expression is induced, while *FBN2* expression is suppressed. *FBN1b* expression is induced by drought but not by cold stress [16]. In response to high light, plants produce abscisic acid (ABA) as a stress signal, and ABA regulates *FBN* expression through ABA-responsive regulators *ABSCISIC ACID-INSENSITIVE 1* (*ABI1*) and *ABSCISIC ACID-INSENSITIVE 2* (*ABI2*). The resulting FBN accumulation enhances photosystem II resistance to photoinhibition [6]. *CDSP34* from tomato was found to be regulated by ABA and high illumination conditions [5,17]. In Arabidopsis, cold/light stress-related jasmonic acid (JA) biosynthesis was conditioned by the accumulation of PG-associated FIB1/2 proteins [10]. Arabidopsis homozygous *fbn5*-knockout mutant line showed the suppressed expressions of genes involved in JA biosynthesis compared to that of the wild type under high light conditions [18]. In rice, overexpressing *OsFBN1* reduced the grain-filling percent and JA levels and increased the formation of PGs, under heat stress in rice [13,19].

ABA is a plant hormone involved in seed germination, dormancy, organ size, stomatal closure, osmotic regulation, and growth inhibition that integrates various stress signals and controls downstream stress responses [20,21]. Plants constantly regulate ABA levels in response to changing environmental conditions [22]. Representative mechanisms for plant responses to environmental stress are ABA-dependent and ABA-independent [23]. In these processes, various transcription factors are involved in the transcriptional activation of stress response genes [22]. Dehydration-responsive element-binding protein 2A/2B (DREB2A/2B), ABSCISIC ACID–RESPONSIVE ELEMENT BINDING PROTEIN1 (AREB1), RD22BP1, MYC/MYB/CRT, ARBE, and MYCRS/MYBRS interact with *cis*-acting elements, such as DRE/CRT, ARBE, and MYCRS/MYBRS, respectively, to regulate expression of stress response genes.

Drought stress mainly causes changes in osmotic pressure between plant cells and the external environment, resulting in dehydration, while salt exposure stresses plant cell osmotic and ionic homeostasis. Drought and salt commonly activate osmotic stress response pathways [24,25,26]. AREB/ABF (ABA-responsive element-binding protein/factor) is an ABA-dependent bZIP transcription factor that interacts with the ABRE (PyACGTGG/TC) region promoters of many ABA-responsive genes conserved in several plant species. Arabidopsis has nine *AREB/ABF* genes. *AREB1/ABF2, AREB2/ABF4,* and *ABF3* are induced by osmotic stress. DREB proteins belonging to the AP2/ERF transcription factor gene family are also associated with drought and osmotic stress resistance [27]. Among these, DREB1A and DREB2A specifically interact with *cis*-acting dehydration-responsive elements (C-repeats) involved in drought and cold stress response gene expression [22,28]. RD29A is induced through an ABA-independent pathway, and RD22 (mediated by ABA) can be activated directly by high salinity and cold stress [28]. Salt and osmotic stress also rapidly activate SnRK2-family protein kinases. Ten SnRK2s, excluding SnRK2.9, are activated by osmotic stress [23,24]. The mechanism by which ABA activates SnRK2 has been identified, but the mechanism by which osmotic stress activates SnRK2 is not yet known. According to a recent study, the B2, B3, and B4 subgroups of Raf-like protein kinase [3] are rapidly activated by osmotic stress, and their activity is required for phosphorylation and activation of SnRK2 [26]. 

This study aimed to investigate the function of FBN11 in the abiotic stress response. FBN11 is the only FBN protein with a kinase domain. FBN11 structural features and intracellular location were analyzed in this study, and *FBN11* expression during an abiotic stress response was analyzed. Differences in stress response between *fbn11* T-DNA-inserted homozygous mutants and wild-type plants under abiotic stress conditions were analyzed. Expression of genes encoding major regulatory transcription factors involved in ABA-dependent and ABA-independent stress response regulatory pathways were analyzed during seed germination under osmotic stress.

## 2. Materials and Methods

### 2.1. Protein Sequence and Phylogenetic Tree Analysis

The protein sequences of the *Arabidopsis thaliana FBN* genes were obtained from TAIR (https://www.arabidopsis.org/ accessed on 24 June 2019) using a previously reported method [29]. These sequences were used to query NCBI BLAST (https://blast.ncbi.nlm.nih.gov/Blast.cgi accessed on 25 June 2019) to search for FBN protein sequences from other plant species. The FBN protein sequences of Chlamydomonas were obtained from the Phytozome12 database (https://phytozome.jgi.doe.gov/pz/portal.html accessed on 28 May 2020). The obtained sequences were aligned using ClustalW with MEGA 7.0 (https://www.megasoftware.net/ accessed on 29 May 2020). The phylogenetic tree was generated using the 1000 bootstrap replication of MEGA 7.0, using the Maximum Likelihood (ML) method.

### 2.2. Subcellular Localization

*FBN11* cDNA lacking its TAG stop codon was amplified by PCR and inserted into *Xba*1 and *Sma*1 sites in the p326-sGFP vector to construct FBN11-sGFP. FBN11-sGFP was transfected with polyethylene glycol (PEG) into protoplasts isolated from leaves of Arabidopsis or *Nicotiana benthamiana* [30]. Transfected protoplasts were incubated in darkness for 22 h at 25 °C. Fluorescence was observed using a confocal laser microscope (Confocal, Leica TCS SP5). GFP fluorescence was excited at 488 nm and detected at 505 at 525 nm. Chlorophyll autofluorescence was exited at 488 nm and detected between 640 and 700 nm.

### 2.3. Plant Material and Growth Conditions

The seeds of Salk_146998C (*fbn11-1*) and Salk_052070C (*fbn11-2*), which are T-DNA-insert *FBN11* gene mutants (AT5G53450), were purchased from the Arabidopsis Biological Resource Center (ABRC: http://abrc.osu.edu/ accessed on 2 March 2016). Arabidopsis wild type (Col-0), and *FBN11* gene mutants *fbn11-1* and *fbn11-2,* were sterilized with 70% ethanol and 0.5% sodium hypochlorite (NaClO). Seeds that had been subjected to imbibition at 4 °C for 3 d were germinated on 1/2 MS (Murashige and Skoog) agar medium containing 1% sucrose and grown to seedlings for 7 d. Seedlings were transferred to a pot with soil and grown at 23 °C under 16 h light/8 h dark conditions. 

### 2.4. T-DNA Insertion Homozygous Mutant Selection

The T-DNA insertion homozygous mutants were selected by analyzing size differences between PCR products amplified with an *FBN11* gene-specific primer and the T-DNA border-specific primer, indicating T-DNA insertion in the gene. Primers for selecting T-DNA insertion mutants were designed using the T-DNA design site (http://signal.salk.edu/tdnaprimers.2.html accessed on 4 March 2017). To select *fbn11-1* (SALK_146998C), LP 5′-TATTAAGGCACGTGTGGAAGG-3′ and RP 5’-GCAACGCTTACAGTACCATGG-3′ were used, and LP 5′-CGAATTTTCAAACCCTAAATCG-3′ and RP 5’ TTGTAAATTCGGCAGATTTGG-3′ were used to select *fbn11-2* (SALK_052070C). The T-DNA left specific primer, LBb1.3 (5′-ATTTTGCCGATTTCGGAAC-3′), was used (Appendix A) to detect T-DNA insertion in the *FBN11* gene.

### 2.5. Seed Germination Test

The germination rates of *fbn11* mutant and wild-type (WT) seeds were assayed under ABA hormone exposure, high salinity, and osmotic stress due to high mannitol exposure [31]. Wild-type and *fbn11* homozygous mutant lines were seeded in 1/2 MS medium without a stressor or in the presence of ABA (1 μM), NaCl (200 mM), or mannitol (300 mM). In the germination experiment, 50 seeds were seeded onto three plates for each experimental group, and the germination rate was measured every 12 h at 23 °C. The percentage of germinated seeds was measured for 7 d after sowing. Germination rates were calculated by dividing the number of germinating seeds (based on when the infiltrated seed coat swelled to reveal young roots) by the total number of seeds.

### 2.6. Primer Design and Construction

Primers used to analyze the expression of the main regulatory transcription factors in the osmotic stress response pathways were prepared using Primer3Plus (http://www.bioinformatics.nl/cgi-bin/primer3plus/primer3plus.cgi accessed on 29 August 2019). Primers for the ABA-dependent/independent pathway-related transcription factor genes *ABF1-4*, *CBF1-3*, *DREB2A*, and *FBN11* were generated based on program selection conditions, with primer lengths of 17–25 bases, 40–60% GC content, and 55–65 °C Tms. To select optimal primer pairs, we confirmed whether sequences were specific to each gene using NCBI BLAST. Primers were synthesized on a 25 nmol scale, and the BioRP purification method was selected when ordering oligos from Bioneer (https://www.bioneer.co.kr/ accessed on 30 August 2019) (Appendix A).

### 2.7. Abiotic Stress Treatments 

Seedlings were grown for 10 d in 1/2 MS medium, then transferred to normal medium without a stressor (1% sucrose + 1/2 MS) as a control, or to a solid medium with 100 μM ABA, 150 mM NaCl, or 300 mM mannitol in 1% sucrose + 1/2 MS for stress testing. At each hour after initiating stress treatments, the root was removed from germinating seeds using scissors, placed in a 2 mL tube, quickly frozen in liquid nitrogen, and stored. Alternatively, 18-day-old seedlings grown in control medium were transferred to 1/2 MS liquid medium containing 1 μM ABA, 150 mM NaCl, or 300 mM mannitol and monitored hourly. At multiple points after initiating treatment, seedling roots were removed with scissors, frozen, and stored as described above.

### 2.8. Measurements of Root Growth

Wild-type and mutant (*fbn11-1* and *fbn11-2*) seedlings grown in 1/2 MS normal medium with 1% sucrose were transferred to a stressor medium containing 20 μM ABA, 200 mM NaCl, or 300 mM mannitol. Plates were incubated for eight days in a 23 °C incubator. Root growth from marked starting points was measured using Image J (https://imagej.nih.gov/ij/ accessed on 24 September 2019).

### 2.9. RNA Isolation

Total RNA was extracted from wild-type and *fbn11* mutants using TRIzol (Favorgen). Plant samples (100 mg) were ground with liquid nitrogen and mixed with 1 mL TRIzol, and allowed to stand at room temperature for 5 min, followed by the addition of 0.2 mL of chloroform. The mixture was left at room temperature for 15 min, then centrifuged at 4 °C for 15 min. After centrifugation, the supernatant was transferred to a 1.5 mL tube, and isopropyl alcohol (IPA) (0.5 mL) was added. RNA pellets were precipitated by centrifugation for 15 min at 4 °C, and pellets were washed with 1 mL of 75% ethanol. Finally, pellets were dried on a clean bench, and 10–30 μL DEPC was added to dissolve each pellet. The dissolved pellets were treated with RNase-free DNase I (Thermo Fisher Scientific, Seoul, South Korea) to remove genomic DNA contamination.

### 2.10. cDNA Synthesis and Quantitative Polymerase Chain Reaction Analyses

cDNA was synthesized from 1 μg of total RNA using the method described below. A total of 10 μL of reaction solution containing 1 μg of total RNA, 125 μM of dNTP, and 2.5 μM of oligo-dT was prepared, incubated at 65 °C for 5 min, and quickly transferred to ice. To this solution, 4 μL of 5X buffer, 20 unit of inhibitor, 200 unit of reverse transcriptase, and 4.5 μL of DEPC were added, and cDNA synthesis was completed at 42 °C for 50 min followed by 95 °C for 5 min using a PCR machine.

cDNA was diluted to 7.1 ng/μL, and real-time qPCR was performed in 96-well blocks using the Step-One Plus RealTime PCR System (Applied Biosystems, Seoul, South Korea) according to the manufacturer’s protocol using PCR primers and SYBR Green Premix (Toyobo, Seoul, South Korea). In this experiment, the housekeeping gene *AteIF4a* (AT3G13920) was used as an internal control [32], and the Ct value of the target gene was normalized. Then, the expression value relative to the control was determined by the 2^−ΔΔCt^ method. All qRT-PCR used total RNA samples extracted from three independent biological replicates. The specificity of qRT-PCR was determined by analyzing the amplified products’ melting curves using standard methods installed in the system.

## 3. Results

### 3.1. Structural Features of FBN11

Analysis of FBN11 protein sequence homology using BLASTP showed that FBN11 possesses a serine/threonine-protein kinase domain spanning amino acids 88 to 399 and an FBN domain spanning amino acids 434 to 587 (a unique feature among other *FBN* gene families) (Figure 1A). The kinase domain was analyzed using SWISS-MODEL (https://swissmodel.expasy.org/ accessed on 2 June 2020), which builds tertiary structure models based on homology to solved structures. Mitogen-activated protein kinase 7 (4ic7.2.A) was the template model with the highest homology (19.62%) among the 50 identified templates (Appendix A). ExPASy prosite (https://prosite.expasy.org/ accessed on 2 June 2020), which can be used to find and characterize domains within proteins, suggests that the kinase domain has two ATP binding sites (involving amino acids 94 and 126, respectively) and one active site (proton acceptor at amino acid position 255) (Figure 1B).

The structure and characteristics of FBN11 protein were analyzed using a program on the protein analysis website (https://web.expasy.org/cgi-bin/protparam/protparam accessed on 2 June 2020). FBN11 was richest in leucine (Leu; 11.3%), followed by serine (Ser) and arginine (Arg) (Appendix A). The secondary structure of FBN11 consists of 32.09% alpha-helix (Hh), 19.25% extended strand (Ee), and 44.33% random coil (Cc) (Appendix A). The isoelectric point (pH at which the protein’s positive and negative charges are equal) was 9.43, and the instability index indicating protein stability in vitro was 49.17. The aliphatic index (an indicator of thermal stability) was 90.34, which indicates high heat stability. The GRAVY value was −0.242, indicating that FBN11 is hydrophobic (Appendix A).

The subcellular location of FBN11 was predicted by TargetP (http://www.cbs.dtu.dk/services/TargetP/ accessed on 2 June 2020 ), which predicted that FBN11 might localize to the chloroplast, with a chloroplast transport peptide sequence value of 0.976. FBN11 analysis using TMMM (http://www.cbs.dtu.dk/services/TMHMM/ accessed on 2 June 2020) gave a spiral crossing membrane value of 0 and determined the number of amino acids in the helix to be 0.03, suggesting that there are no transmembrane segments.

Summarizing the above results, FBN11 is a soluble hydrophobic FBN domain-containing protein with a kinase domain fused to the N-terminal region and a predicted chloroplast localization sequence.

### 3.2. FBN11 Subcellular Location

FBNs have been reported to be present in chloroplasts [33]. To visualize subcellular localization of FBN11, a vector containing FBN11-sGFP under CaMV 35S promoter control was introduced into Arabidopsis and tobacco protoplasts, and sGFP green fluorescence was observed using fluorescence and confocal fluorescence microscopy, respectively. The small circular dense green fluorescent light of FBN11-sGFP overlaps with chlorophyll autofluorescence in the chloroplasts of Arabidopsis and tobacco. The location of FBN11-GFP in chloroplasts in two different plant species suggested that wild-type FBN11 may also exist in chloroplasts (Figure 2).

### 3.3. Evolution of FBN11

FBN11 orthologs were retrieved from the genome database of 14 dicotyledons, 5 monocotyledonous higher plants, *Selaginella moellendorffii* [(spike moss), which is representative lycophytes], *Physcomitrella patens* and *Marchantia polymorpha* (bryophytes), *Chara braunii* (charophytes), and *Chlamydomonas reinhardtii* (green algae) (Appendix A). A phylogenetic tree was created to investigate the evolution of FBN11 in land plants (Figure 3). The tree indicated that FBN11 of lycophytes existed in the same clade (88–48% identity with AtFBN11) with FBN11s of higher plants. In addition, FBN11 orthologs (47–36% identity with AtFBN11) of *Chara braunii* and *Marchantia polymorpha* were distributed into the clade distinct from FBN11 orthologs of spike moss. STT7 of green algae *Chlamydomonas reinhardtii* and the bryophytes *Physcomotrella patens* have only a region similar to the kinase of FBN11 of higher plants. The above analysis suggests the existence of FBN11 in the higher plant after lycophytes during evolution.

Comparing the FBN gene family between Arabidopsis and the green algae Chlamydomonas revealed a total of 10 FBNs (PLAP) of Chlamydomonas showing homology with Arabidopsis (Appendix A). After amino acid sequence analysis, the kinase domain of Arabidopsis FBN11 showed 34% homology with the kinase domain of CreSTT7 from Chlamydomonas and 26% homology with CrePLAP2 from Chlamydomonas (Figure 4, Appendix A). Analysis of FBN11 orthologs between higher plants and single-celled green algae suggests the possibility that kinase and FBN genes existed independently in the single-celled alga Chlamydomonas but were conjoined in FBN11 higher plants via a form of fusion by domain shuffling during evolution (Figure 4, Appendix A).

### 3.4. FBN11 Expression Profiles in Developing Tissue ± ABA, NaCl, or Mannitol Exposure

*FBN11* expression in various tissues ± abiotic stress conditions was investigated using data from the TAIR eFP browser (http://bar.utoronto.ca/efp2/Arabidopsis/Arabidopsis_eFPBrowser2.html accessed on 7 June 2019). *FBN11* was expressed in all tissues, including leaves, stems, flowers, roots, and siliques (Appendix A). Among the five tissue types, expression was highest in the leaves, followed by flower, silique, stem, and root, in descending order (Appendix A). *FBN11* expression increased at 30 min, 1 h, and 3 h in 7-day-old wild-type seedlings treated with 10 μM ABA (Appendix A). When 18-day-old seedlings were treated with 150 mM NaCl, *FBN11* expression increased significantly at 3 h and 6 h, decreased at 12 h, and increased again at 24 h (Appendix A). When 18-day-old seedlings were treated with 300 mM mannitol, the results were similar to those for treatment with 150 mM NaCl (Appendix A).

To confirm that the results of the above analyses are reproducible, total RNA was extracted from each stressed tissue type from 10- and 18-day old seedlings for RT-qPCR analysis *of FBN11* expression. When seedlings were treated with 100 μM ABA, *FBN11* expression was elevated at 6 h, 12 h, and 24 h. When seedlings were treated with 150 mM NaCl, *FBN11* expression remained unchanged for up to 6 h, then declined at 12 h. In tissues treated with 300 mM mannitol, *FBN11* expression increased significantly at 6 h, decreased at 12 h, then increased again at 24 h compared to untreated tissue (Figure 5A). In order to investigate the expression pattern of *FBN11* at low concentrations of ABA, 1 μM ABA was treated for 18-days-old seedlings. Treatment with 1 μM ABA did not induce *FBN11* expression. Expression was slightly reduced at 6 and 12 h in 18-days-old seedlings. Exposure to 150 mM NaCl tended to decrease *FBN11* expression compared to non-treated control tissues. When seedlings were treated with 300 mM mannitol, *FBN11* expression was significantly increased at 3, 6, and 24 h (Figure 5B). When wild-type plants were treated with 150 mM NaCl, the expression of *FBN11* was decreased at 6 h (Figure 5B), but a contrast pattern appears in the Arabidopsis gene expression atlas (eFP browser) under the same conditions that showed an induced expression (Appendix A). The difference between these two experiments cannot rule out the possibility that the expression of the *AteIF4a* internal control gene was affected by NaCl treatment. In both experiments, *FBN11* expression was significantly and rapidly increased by treatment with 300 mM mannitol. These results suggest that *FBN11* is induced by osmotic changes and functions as an important osmotic stress response gene.

### 3.5. Characterization of fbn11 Mutants

To characterize FBN11′s possible role in abiotic stress tolerance, *fbn11* homozygous loss of function mutants (due to T-DNA insertion into *FBN11* genomic DNA) were identified. Using *FBN11* gene-specific primers and a T-DNA border primer, homozygous mutants were selected from the salk_052070C and salk_146998C lines based on differences in PCR product size. Two types of *fbn11*homozygous mutants were selected. As shown in the graphical representation of the *FBN11* gene, one has *fbn11-2* with T-DNA inserted in the 5’-UTR region, and the other has *fbn11-1* with T-DNA inserted in the 9th exon (Figure 6A). Genomic DNA was isolated from wild-type and mutant leaves, and PCR was performed using a T-DNA left border targeting primer (LBb1.3) and an *FBN11* gene-specific primer pair (LP and RP primers for T-DNA orientation). As shown in Figure 6B,C, 1130 and 1240 bp bands were found in the wild type, and smaller bands were found in the *fbn11-1* and *fbn11-2* mutants. RT-PCR results confirmed the loss of *FBN11* transcripts in both homozygous mutants (Figure 6D).

### 3.6. Phenotype Comparison between Wild Type and fbn11 Mutants Exposed to Abiotic Stresses

To elucidate FBN11 function in abiotic stress, phenotypic differences in drought stress were investigated using wild-type, *fbn11-1,* and *fbn11-2* mutants. The rate of water loss is an early indicator of drought stress tolerance. According to the water loss measurement method proposed by Baek et al. [34], four-week-old wild-type and mutant shoots were cut from the roots, separated, and weighed immediately. They were placed at room temperature in a covered plate and reweighed periodically to determine the decrease rate of weight (Figure 7A). To verify these results by an alternative method, three-week-old wild-type and mutant leaves of the same developmental period were separated from the plant and placed in a dry environment using a hood, and the rate of water loss was again measured (Figure 7B). There was no more than a 2% difference in water loss rate between wild type and fbn11 mutant plants in both cases.

To determine whether the *fbn11* mutant displayed a difference in root development, a 5-day-old seedling was transferred from 1/2 MS medium to medium containing 20 μM ABA, 200 mM NaCl, or 300 mM mannitol, and root growth and development was observed for eight days (Figure 7C). The distance that the root tips extended over eight days was measured using Image J software (https://imagej.nih.gov/ij/ accessed on 24 September 2019). Root length in normal medium extended by 23.2 mm in wt, 23.9 mm in *fbn11-1*, and 24.7 mm in *fbn11-2*. Average root growth in 20 μM ABA medium was 12.5 mm (wt), 10.6 mm (*fbn11-1*), and 15.8 mm (*fbn11-2*). Average growth in 200 mM NaCl medium was 1.08 mm (wt), 1.54 mm (*fbn11-1*), and 1.13 mm (*fbn11-2*). Average growth in 300 mM mannitol was 8.49 mm (wt), 6.55 mm (*fbn11-1*), and 7.07 mm (*fbn11-2*). In all cases, no distinct difference was found between wild type and mutant (Figure 7D). Taken together, loss of FBN11 function did not significantly impact plant response to ABA, NaCl, or mannitol treatment.

### 3.7. Analysis of fbn11 Effect on Seed Germination Pattern under Different Stress Conditions

Expression profiles of 14 *FBN* family genes in the late stage of seed maturation revealed the strongest *FBN11* expression (Appendix A). These results suggest a possible role for FBN11 in seed germination.

The germination rates of wild type and *fbn11* homozygous mutant seeds were compared under stressed conditions. In standard 1/2 MS medium with no stress treatment, germination of *fbn11* mutant seeds was 48 h slower than wild type. A 100% germination rate was observed at 72 h. At that point, there was no difference between wild type and mutant germination rates (Figure 8A). The germination rate of *fbn11* mutants in 1/2 MS medium supplemented with 1 μM ABA decreased after 96 h compared to that of the wild type (Figure 8B). In the medium supplemented with 200 mM NaCl, seed germination rates of both wild type and *fbn11* mutants were slow for 48 h, and the difference in germination rates between wild-type and *fbn11* mutants increased after 72 h, suggesting that the germination rate of *fbn11* was lower than that of the wild type (Figure 8C). In the medium containing 300 mM mannitol, a clear difference in germination rates of wild-type and *fbn11* mutants began to be apparent by 48 h. Under these conditions, the wild-type germination rate increased sharply at 72 h, whereas *fbn11* mutant seed germination remained delayed (Figure 8D). Figure 7E,F show the final germination and green cotyledon formation rates of wild type and *fbn11* mutant seeds after 7 d in ABA, NaCl, or mannitol supplemented medium. Compared to wild type, the *fbn11* mutant germination rate was decreased in stress-inducing media. The germination rate of *fbn11-2* in NaCl and mannitol medium was significantly lower than that of the *fbn11-1* mutant and the wild type (Figure 8E). In addition, the green cotyledon formation rate was significantly lower in *fbn11* mutants, especially in mannitol, compared to wild type (Figure 8F and Appendix A). In osmotic stress, *fbn11-2* with T-DNA inserted in the first exon of 5’-UTR showed a stronger seed germination inhibition phenotype than *fbn11-1* inserted in the tenth exon (Figure 6, Figure 8). This phenomenon suggests the possibility that truncated FBN11 produced by T-DNA insertion in *fbn11-1* may play an incomplete role (Appendix A). Taken together, these results indicate that FBN11 responds sensitively to mannitol-induced osmotic stress.

### 3.8. Expression of Abiotic Stress-Marker Transcription Factors in fbn11 during Seed Germination under Osmotic Stress

When wild-type seedlings were treated with osmotic stress, or mannitol, *FBN11* expression increased (Figure 5), and seed germination and cotyledon formation rates were decreased in *fbn11* mutants compared with wild type (Figure 8E,F). Since this result suggests that FBN11 present in chloroplasts may regulate the expression of genes involved in the osmotic stress response, we assessed by RT-qPCR whether expression of transcription factors involved in ABA-dependent/independent pathways of abiotic stress response changes (Figure 9). The seeds used in these experiments were directly germinated for 116 h on a solid medium containing 300 mM mannitol, at which point, wild-type seeds were all germinated, and germinated and non-germinated seeds of *fbn11* mutants were collected. Total RNA was extracted. In ABF1-4, a representative transcription factor family in the ABA-dependent pathway, when osmotic stress was applied to wild-type seed germination, *ABF1* and *ABF3* increased, but *ABF2* and *ABF4* did not change significantly. By contrast, *fbn11* mutants showed a decrease in *ABF1-4* expression during seed germination compared to wild type. Considering that no significant difference in *ABF1-4* expression is seen between wild-type and *fbn11* mutants under normal conditions, decreased expression in *fbn11* knockouts suggests that FBN11 regulates *ABF1-4* expression under osmotic stress (Figure 9A).

Changes in expression of *DREB2A* and *CBF1-3*, representative transcription factors in the ABA-independent pathway, were also investigated during germination of wild-type and *fbn11* mutant seeds under osmotic stress. Expression of *DREB2A* was significantly increased in both wild-type and *fbn11* mutants by treatment with mannitol, but no difference in expression was observed between wild-type and the *fbn11* mutants. *CBF2* and *CBF3* showed a tendency to increase during osmotic stress in wild type slightly, but the expression in *fbn11* mutants was similar to that of wild type under normal and osmotic stress conditions. By contrast, *CBF1* (CRT/DRE-binding factor 1, also named *DREB1B*) expression significantly increased during osmotic stress in the wild type but decreased significantly in the *fbn11* mutants. These results suggest that in response to stress, FBN11 present in chloroplasts may affect the expression of different stress response modulators (Figure 9B).

## 4. Discussion

FBN11 is a unique FBN paralog with a kinase in its N-terminal region, unlike FBNs found in higher plants (Figure 1). The primary structure of the FBN11 kinase domain revealed probable tertiary structural similarity (19.62%) with human mitogen-activated protein kinase 7 (MAPK7) (tertiary molecular model: 4ic7.1.A). MAPK is involved in biotic and abiotic stress responses in plants [35]. The characteristic of FBN11 having FBN domains that have similarities with MAP Kinase suggests that FBN11 may be involved in recognition and response to stress in chloroplasts. Whether FBN11 kinase plays a role in chloroplasts and which chloroplast protein(s) is/are its cognate substrate(s) should be the subjects of future studies.

FBN11 exists only in higher plants. In this study, we identified independent genes homologous to the kinase or FBN domains in the green algae Chlamydomonas genome database. Our results suggest the possibility that FBN11 was generated by domain shuffling, resulting in a kinases/FBN fusion during the evolution of higher land plants from green algae (Figure 4). The conserved presence of FBN11 in higher plants suggests the existence of a defense mechanism involving signal transduction in chloroplasts in response to biotic and abiotic stresses.

FBN proteins have an FBN domain, which is a hydrophobic region with the potential to bind fat-soluble molecules. FBN1 in red pepper binds to carotenoids to form microfibril structures [2]. The FBN domain of FBN5 can help maintain the activity of solanesyl diphosphate synthase by binding solanesyl diphosphate, a hydrophobic tail structure of plastoquinone involved in photosynthetic electron transport [36]. A possible function of the FBN domain of FBN11 may be an induction of kinase domain activity by binding to an as yet unidentified hydrophobic substrate.

All *FBN* family genes discovered to date are present in chloroplasts or plastids. Most of them are present in plastoglobules (plastid localized lipoprotein particles) [15]. Some, such as FBN5, are present in the chloroplast stroma [36]. FBN11 has been confirmed to be present in chloroplasts (Figure 2). It has been predicted, based on its isoelectric point and Ref. [6] and hydrophobicity, to exist in the thylakoid. FBN11 has no transmembrane domain and is weakly hydrophobic. Although we have confirmed that FBN11 is present in the chloroplast, its precise localization within the chloroplast remains elusive.

*FBN11* gene expression increases under osmotic stress [23] (Figure 5 and Appendix A), and seed germination and cotyledon formation were adversely affected under osmotic stress conditions in *fbn11* mutants (Figure 8 and Appendix A). These results suggest that FBN11 mediates a regulatory response to osmotic stress. Transcription factors involved in ABA-dependent and ABA-independent pathways have been reported to regulate gene expression in response to osmotic stress [23,28]. The ABA-dependent signaling pathway activates and regulates the ABF/ABRE transcription factor through multisite phosphorylation of conserved domains by SNF1-related kinase2 (SnRK2s) upon osmotic stress [37,38,39]. The ABA-independent pathway involves induction of DREB2A protein by drought and high salt concentrations. DREB2A specifically binds to the DNA sequence 5’-[AG]CCGAC-3’ and interacts with ABF/AREB/factors. Transcription factors of the ABA-dependent pathway regulate stress response genes or interact with DRE/CRT *cis*-elements to express downstream genes involved in drought and salt stress responses [40].

In this experiment, four types of ABA-dependent transcriptional regulators (*ABF1-4*) showed no significant change in gene expression under osmotic stress during seed germination in wild-type but displayed significantly decreased transcripts in *fbn11* mutants relative to wild-type (Figure 9A). Among the four transcriptional regulators of the ABA-independent pathway, only *DREB2A* expression increased significantly in wild-type and *fbn11* mutant germinating seeds under osmotic stress. Under osmotic stress during seed germination, *CBF1* expression decreased in the *fbn11* mutants (Figure 9B). These results suggest that FBN11 present in chloroplasts responds to osmotic stress by regulating *ABF1-4* expression in ABA-dependent and *CBF1* in ABA-independent pathways in the nucleus by retrograde signaling.

Plants detect biotic or abiotic environmental stresses via organelles within their cells and respond with changes in gene expression and/or metabolic or physiological changes. The signal-sensing organelles (endoplasmic reticulum (ER) and chloroplasts) experience protein misfolding or accumulation of unfolded protein during environmental stresses. Such stresses are detected by a specific sensor protein in the ER membrane [41]. Chloroplasts are organelles in which many metabolic reactions, including photosynthesis, occur, and metabolic balance is easily disturbed by environmental changes. These disturbances are transmitted to the nucleus through retrograde signals, allowing chloroplasts to regulate cellular activity [42,43,44].

This study raises many unanswered questions but supports the following hypothesis (diagrammed in Figure 10). When subjected to osmotic stress during seed germination, the chloroplast recognizes this osmotic stress, producing an unknown lipid molecule that acts as a secondary signal transducer by binding to the FBN domain of FBN11, thereby activating the kinase domain to transduce a retrograde signal. This retrograde signal responds to osmotic stress by regulating the expression of *ABF1-4* and *CBF1* transcription factors in the nucleus (Figure 10). The expression of *FBN11* is predicted to be regulated by the OBP3, a Dof transcription factor, which responds to salicylic acid (SA). On the contrary, SA treatment upregulates *FBN11* expression, and jasmonate treatment downregulates *FBN11* [45]. ABF1 plays an essential role in the salt/osmotic stress-resistant germination phenotype by suppressing a negative regulator of light development, *DEETIOLATED 1* (*DET1*) [46]. ABF1 is also activated by SNF1-related kinase 2s (SnRK2s) in response to osmotic stress during vegetative growth [27]. CBF1 directly controls the expression of two Arabidopsis glycosyltransferase genes, *UGT79B2* and *UGT79B3*, which confer salt stress tolerance via modulating anthocyanin accumulation [47]. These results suggest that FBN11 may be involved in disease resistance as well as osmotic stress.

## Figures and Tables

**Figure 1 biology-10-00368-f001:**
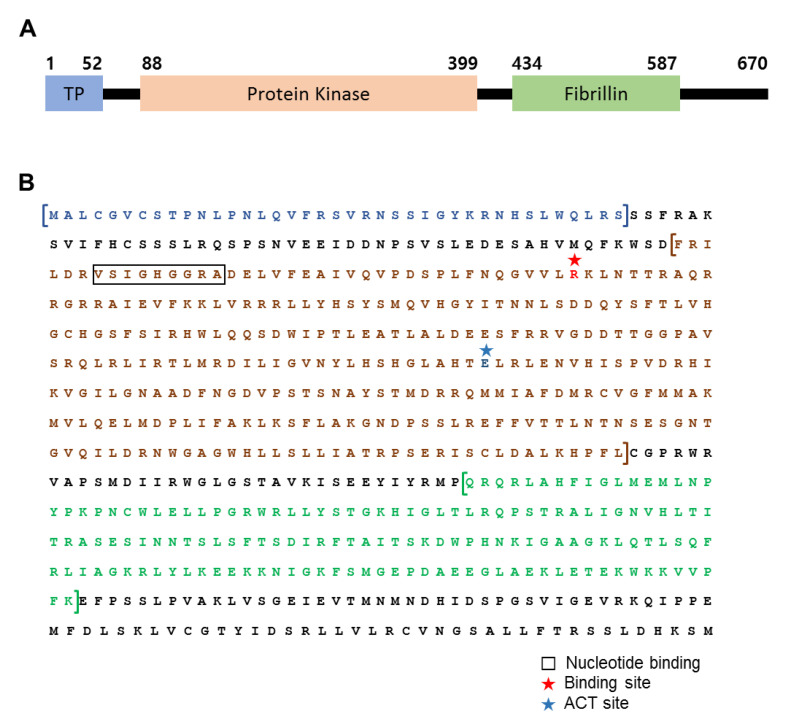
Arabidopsis FBN11 protein structural characteristics. (**A**) Schematic of FBN11 domain structure with positions labeled in amino acid numbering. 1–52; transit peptide, 88–399; Protein kinase domain (serine/threonine-protein kinase), 434–587; Fibrillin conserved domain. (**B**) The amino acid sequence of FBN11. Blue brackets indicate TP, brown brackets indicate the kinase domain, and green brackets indicate the fibrillin domain. The red star indicates a nucleotide-binding site predicted to bind nucleotide phosphates. A binding site describes an interaction between a single amino acid and another chemical entity. The blue star indicates a catalytic residue in the kinase active site (ACT).

**Figure 2 biology-10-00368-f002:**
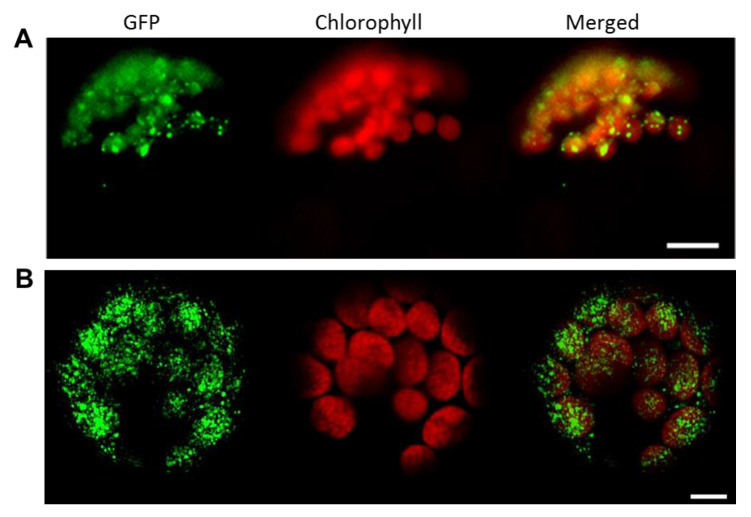
Localization of FBN11-GFP fusion protein to *Arabidopsis thaliana* (**A**) and *Nicotiana benthamiana* (**B**) protoplasts. GFP = green fluorescence protein imaging; Chlorophyll = chloroplast red auto fluorescence; Merged = merged GFP and chloroplast fluorescent images. Scale bars = 20 μm (*A. thaliana* ) and 5 μm (*N. benthamiana*).

**Figure 3 biology-10-00368-f003:**
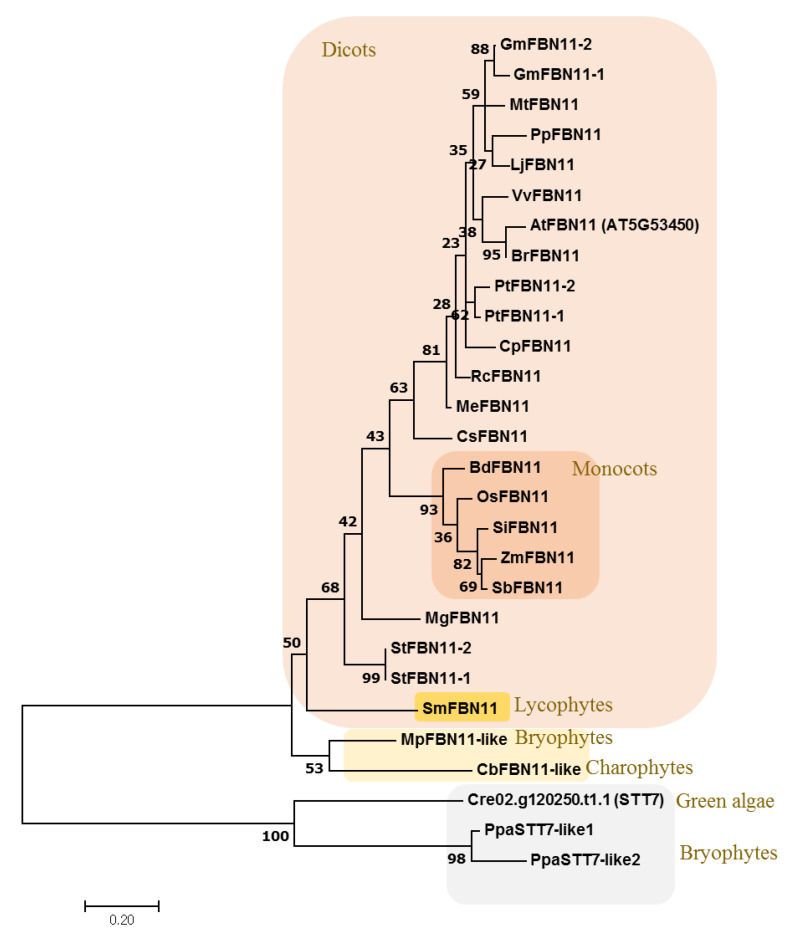
The phylogenetic tree between FBN11 orthologs from dicot (14 species), monocots (5 species), lycophytes (1 species), bryophytes (2 species), charophytes (1 species), and chlorophyte (1 species). The tree was analyzed using the Maximum Likelihood (ML) method. Bootstrap values are given at the nodes as a percentage of 1000 replicates. The scale bar indicates the number of amino acid substitutions per site. At: *Arabidopsis thaliana*, Br: *Brassica rapa*, Cp: *Carica papaya*, Cs: *Cucumis sativus*, Gm: *Glycine max*, Lj: *Lotus japonicas*, Me: *Manihot esculenta*, Mg: *Mimulus guttatus*, Mt: *Medicago truncatula*, Rc: *Ricinus communis*, Pp: *Prunus persica*, Pt: *Populus trichocarpa*, St: *Solanum tuberosum*, Vv: *Vitis vinifera*, Bd: *Brachypodium distachyon*, Os: *Oryza sativa*, Sb: *Sorghum bicolor*, Si: *Setaria italic*, Zm: *Zea mays*, Sm: *Selaginella moellendorffii*, Ppa: *Physcomitrella patens*, Mp: *Marchantia polymorpha*, Cb: *Chara braunii*, Cre: *Chlamydomonas reinhardtii*. Information for these genes is in Appendix A.

**Figure 4 biology-10-00368-f004:**
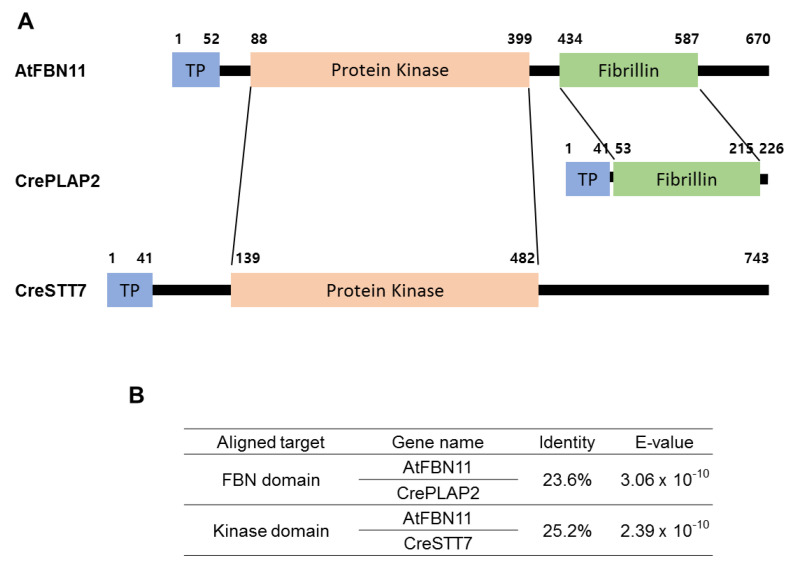
Comparison of domain structure and homology between Arabidopsis FBN11 and *C.reinhardtii* PLAP2 and STT7. (**A**) FBN11 has a transit peptide in aa 1–51, a protein kinase domain at aa 88–199, and an FBN domain at aa 434–587. CrePLAT2 has a transit peptide in aa 1–44 and a second transit peptide in aa 53–215. CreSTT7 has a transit peptide in aa 1–41 and a protein kinase domain at aa 139–482. (**B**) Amino acid sequence identity between the kinase and FBN domains of Arabidopsis FBN11 and the corresponding kinase domain of CreSTT7 and FBN domain of CrePLAP2, respectively.

**Figure 5 biology-10-00368-f005:**
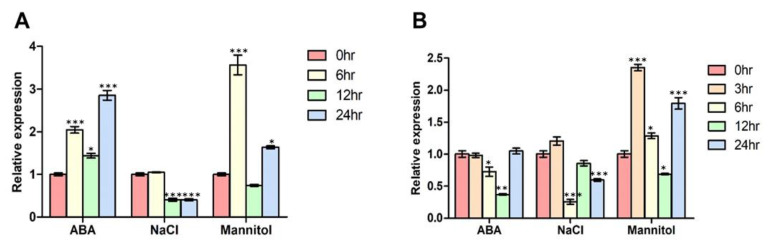
FBN11 expression in Arabidopsis seedlings in response to ABA, NaCl, or mannitol. (**A**) Ten-days-oldArabidopsis seedlings were treated with 100 μM ABA, 150 mM NaCl, or 300 mM mannitol for 0, 6, 12, and 24 h in solid MS medium. (**B**) Eighteen-day-old Arabidopsis seedlings were treated with 1 μM ABA, 150 mM NaCl, or 300 mM mannitol for 0, 3, 6, 12, and 24 h in the liquid medium. *FBN11* expression in (**A**,**B**) is normalized with housekeeping gene *AteIF4a* expression and compared with the non-treated control seedlings. Statistical analysis is by one-way ANOVA with Tukey’s multiple comparisons test (* *p* < 0.05, ** *p* < 0.01, *** *p* < 0.001).

**Figure 6 biology-10-00368-f006:**
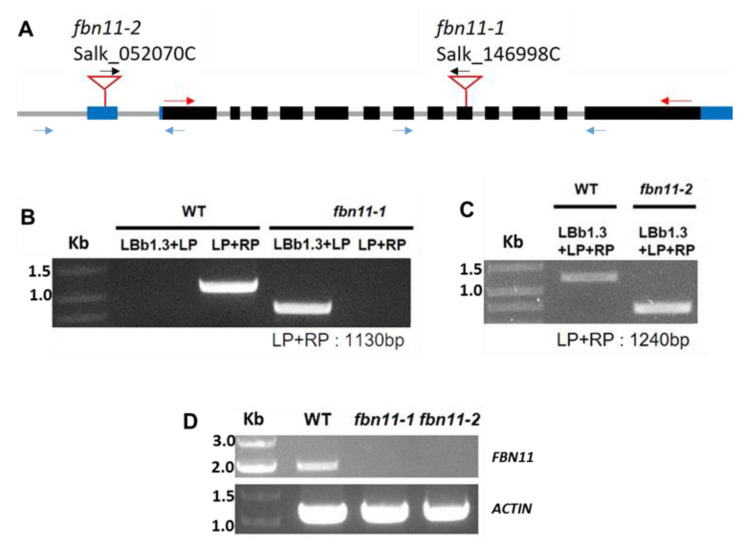
Identification of *fbn11-1* and *fbn11-2* homozygous mutants containing T-DNA inserts. (**A**) Schematic diagrams of mutant *FBN11* genes with T-DNA insertions. Black boxes, grey lines, and blue boxes represent exons, introns, and untranslated regions, respectively. (**B**,**C**) Genotyping of wild-type, *fbn11-1*, and *fbn11-2* homozygous mutant plants by genomic DNA PCR. PCR primer positions are shown by arrows (Blue for LP and RP, Red for RT-PCR). Expected PCR product sizes for T-DNA detection in genes amplified with the primer pairs 11-1LP and 11-1RP (1130 bp), 11-2LP, and 11-2RP (1240 bp) are indicated in B and C. (**D**) RNA transcript levels of *FBN11* genes in wild type (WT) and homozygous *fbn11-1*, and *fbn11-2* mutant seedlings. RT-PCR of total RNAs isolated from 3 week old Arabidopsis plant. *ACTIN* was used as a loading control.

**Figure 7 biology-10-00368-f007:**
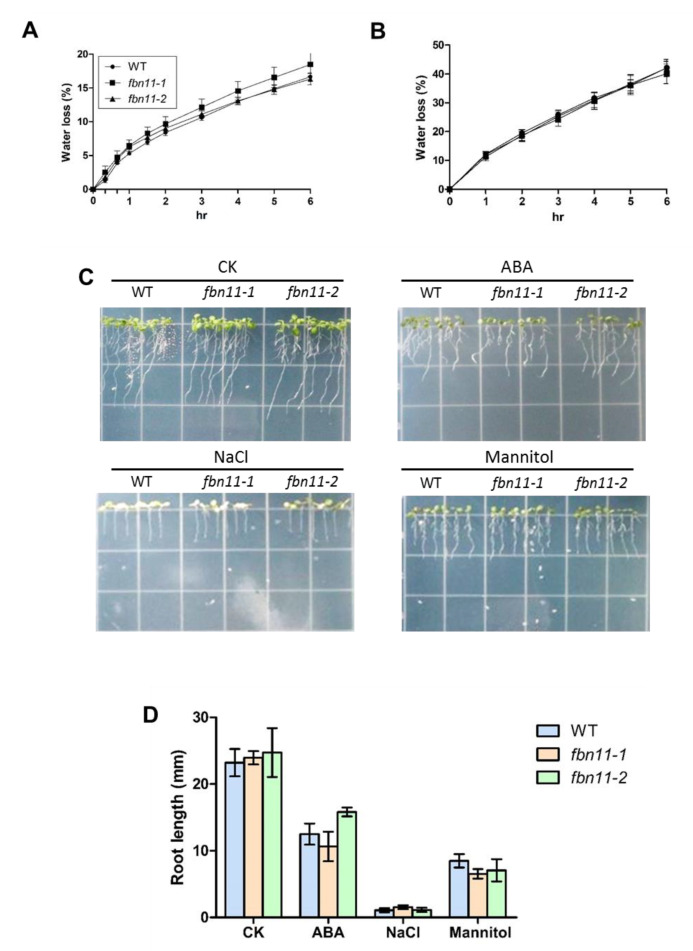
Measurement of water loss due to drought stress conditions and effect on root growth. (**A**) Four-week-old plant shoots were detached from the root and weighed immediately. Shoots were then placed in covered plates at room temperature and weighed at various time intervals. Loss of fresh weight was calculated based on the initial shoot weight. At least three biological replicates were measured for each sample. (**B**) Rates of water loss by leaves were also measured. Short-term assays were performed with detached leaves of matching developmental stage and size from 21-day-old plants. Five leaves per individual were excised, weighed fresh, subjected to the drying atmosphere of a flow laminar hood, and weighed at fixed time points while drying. Kinetic analyses of water loss were performed and are represented as percentages of initial fresh weight at each time point. (**A**,**B**) There were no statistically significant differences (*p* > 0.05) in water loss rates between WT and *fbn11* mutants at any time point (one-way ANOVA with Tukey’s multiple comparisons test). (**C**) Root length phenotype and (**D**) measurement of root growth under stress. CK, control media; ABA, supplemented with 20 μM ABA; NaCl, supplemented with 200 mM NaCl; mannitol, supplemented with 300 mM mannitol. There were no statistically significant differences (*p* > 0.05) in root length between the stress treatment groups at any time point (one-way ANOVA with Tukey’ s multiple comparisons test).

**Figure 8 biology-10-00368-f008:**
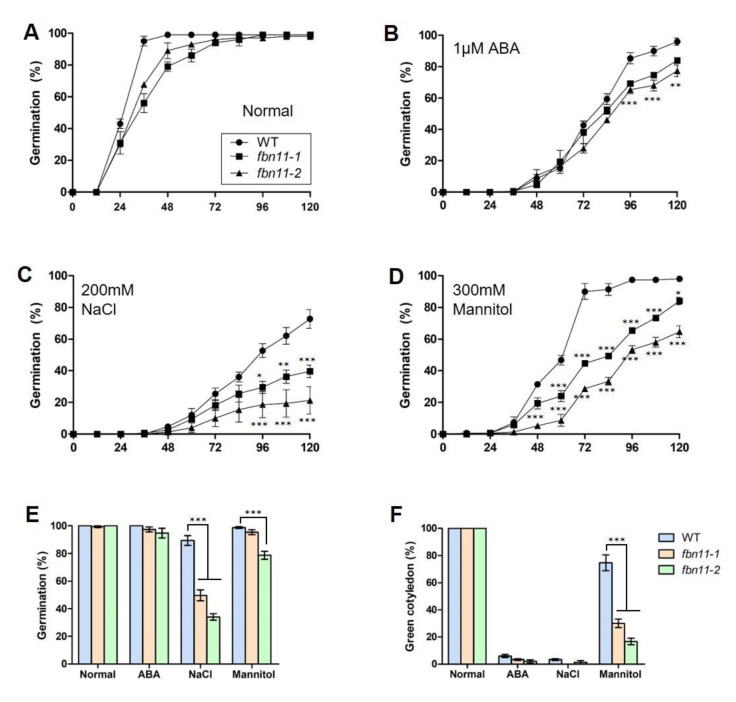
Seed germination and cotyledon formation of wild type and *fbn11* homozygous mutants exposed to ABA, NaCl, and mannitol. (**A**) Seed germination under normal conditions. or (**B**) supplemented with 1 μM ABA, (**C**) 200 mM NaCl, or (**D**), 300 mM mannitol. (**E**,**F**) represent percentages of seed germination and green cotyledon development in ABA, NaCl, or mannitol stressed conditions. Approximately 50 seeds of each genotype were sown on each plate and scored for germination and early growth 7 d later. Each value corresponds to mean ± standard deviation (SD). Statistical analysis is by one-way ANOVA with Tukey’s multiple comparisons tests (* *p* < 0.05, ** *p* < 0.01, *** *p* < 0.001). Normal; control media, ABA; supplemented with 1 μM ABA, NaCl; supplemented with 200 mM NaCl, mannitol; supplemented with 300 mM mannitol.

**Figure 9 biology-10-00368-f009:**
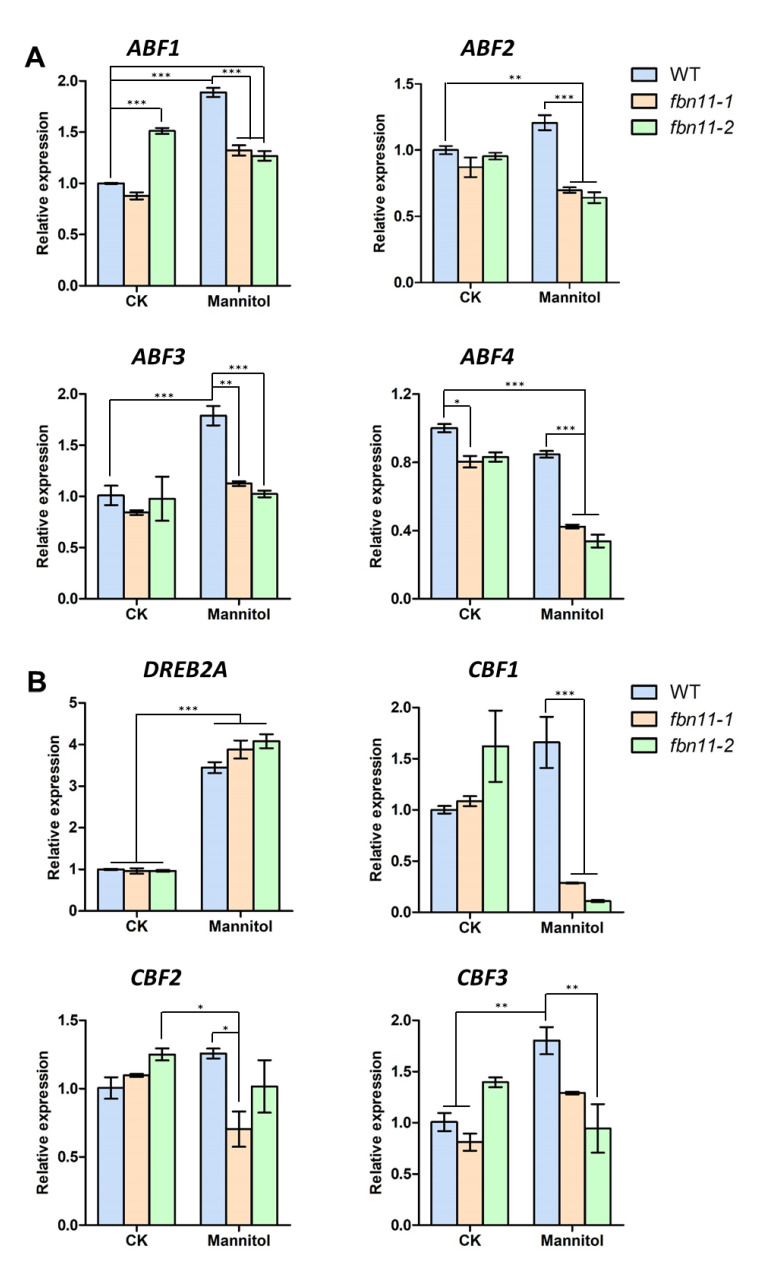
Expression of key transcription factors mediating ABA-dependent/independent pathway genes during seed germination in wild type and *fbn11* homozygous mutants. (**A**) Transcription factors in the ABA-dependent pathway (**B**) Transcription factors in the ABA-independent pathway. Total RNA was extracted from seeds undergoing germination after 116 h treatment with 0 or 300 mM mannitol. To determine the internal control for qRT-PCR analysis, the unaffected expression pattern of the *AtelF4* gene was checked in wild-type and *fbn11* mutant seed germination under mannitol treatment conditions (Appendix A). Statistical analysis is by one-way ANOVA with Tukey’s multiple comparisons tests (* *p* < 0.05, ** *p* < 0.01, *** *p* < 0.001).

**Figure 10 biology-10-00368-f010:**
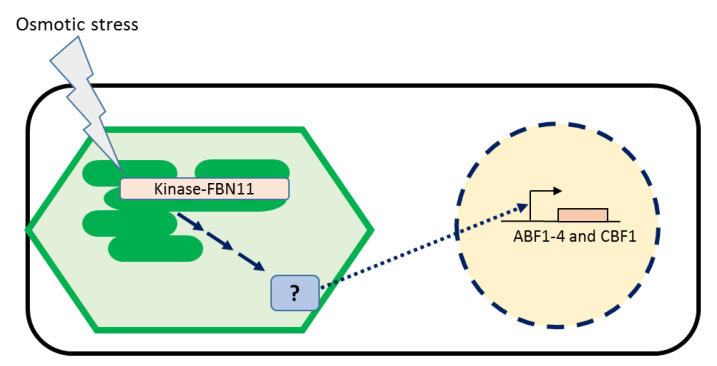
Proposed model for FBN11 function within the osmotic stress response.

## Data Availability

Not applicable.

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
