# Peer review of "Chloroplast Localized FIBRILLIN11 Is Involved in the Osmotic Stress Response during Arabidopsis Seed Germination"

_biology, 2021, doi:10.3390/biology10050368_

Round 1
Reviewer 1 Report
The manuscript by Yu-Ri Choi and colleagues studies an interesting gene, FBN11, which is involved in chloroplast signalling pathway, using two Arabidopsis mutants and adequate experimental plan. They provide new elements regarding this aspect and the manuscript is well-written with supportive figures and tables. However, I have two main concerns related to gene expression analysis:
- the authors are not using the gene specific amplification efficiency, which should be performed prior to the gene expression analysis with serial dilution of pooled cDNA, and further used for quantification with the delta delta CT method
- the authors normalise the expression of target genes using a single reference gene, eIf4A. There is no proof that this gene is stably expressed between all treatments, and I believe that a single gene will not be able to normalise the data with all the assessed abiotic treatments at different time points.
These two points have their importance, since most of the fold-change reported in this manuscript are not very high. In addition, MDPI requests that the analyses should be performed with the highest technical standards.
I have also other, less important remarks that should be corrected before publication:
- Line 186: please correct the title
- In Figure S4, please define the error bars and perform statistics to highlight significant differences. Correct the legend.
- L. 205: the authors should explain why they decreased ABA concentration from 100 µM to 1 µM in 18-day-old seedlings.
- L. 239: please correct to fbn11-2.
- L. 290: Before such a statement, the authors should guarantee that the gene expression levels provided in BAR allow gene-to-gene comparison, and not merely tissue-to-tissue comparison for a given gene. Then, assuming that this gene-to-gene comparison is possible, FBN11 does not appear to be the unique FBN highly expressed during late-stage seed maturation. FBN1 to FBN4 are also highly expressed. The authors should therefore correct this statement.
- L. 355: the authors should be more careful about the modulation of CBF1 in fbn11 mutants. I think this is unlikely that FBN11 directly modulates CBF1, because FBN11 is not a transcription factor. The authors should instead state that the deficiency in FBN11 leads to a wide plant response, and the up-regulation of CBF1 is part of this response.
- L. 416: The authors may describe in further details the functions of ABF1-4 and CBF1 in the framework of retrograde signalling, for instance by emphasising the genes regulated by these two transcription factors and the resulting adaptation of plant physiology, especially aspects related the chloroplasts. Furthermore, FBN11 is aka as ORG1, thus being regulated by the Dof transcription factor OBP3. The authors should elaborate around this point and check whether this may be part of this retrograde signalling.
- L. 457: please precise the settings of the microscope.
Author Response
Please see the attached PDF for "response to reviewer1"
Thank you.

Reviewer 2 Report
In their manuscript, Choi et al. set out to characterize an unusual gene, FIBRILLIN11, that encodes an evolutionarily-fused serine/threonine protein kinase and fibrillin domain that putatively localizes to chloroplasts. The manuscript is straightforward to read and experiments are, for the most part, clearly explained and designed using classical, “tried-and-true” approaches.
The only major mention of this gene in the literature to date has been as a target of salicylic-acid responsive Dof transcription factor OBP3; this paper is not cited here, however, despite the important role of SA in promoting stress responses. I think there should be some discussion of this, and that the authors could potentially use SA as a “positive control” condition for induction of FBN11 in the RT-qPCR experiments (more on those experiments below)
The subcellular localization experiment is fine, but without demonstrating genetic complementation, it is not clear whether these fusion proteins are functional, which is critical for validating localization (not fully sufficient, but necessary). If that experiment is beyond the scope of what the authors wish to show here, then the localization should be clearly described as “putative” throughout. The figure legend (Fig. 2) is technically correct, but it would be nice to more clearly acknowledge that this might not fully reflect wild-type localization. Complementation experiments, incidentally, would be especially useful to demonstrate that fbn11-2 and fbn11-1 phenotypes are due to loss of fbn11 and not due to other effects (more below).
The evolutionary analysis is incomplete, because it does not help to resolve when this chimeric gene (a fusion of a S/T kinase and a FBN) occurred. Monocots and dicots diverged some ~150-200 Mya; Chlamydomonas diverged well over 1 Bya. That leaves over 800 My unexplored! A quick BLAST search suggested that this gene probably was at least present as far back as lycophytes (represented by Selaginella moellendorfii), but may not be conserved in bryophytes (represented by Physcomitrium patens and Marchantia polymorpha), and might have already existed in charophytes (represented by Chara braunii). I recognize that a full phylogenetic treatment is beyond the scope of this report, but at least some mention of the Selaginella orthologues would be useful if the evolution of this gene is going to be discussed, and some further investigation about the earliest appearance of the FBN11-like chimera genes might increase impact of this report.
The use of eIF4a as a reference gene for RT-qPCR is problematic. Although eIF4a might be a “housekeeping gene”, its expression is very dynamic in response to various cues. 150 mM NaCl and 300 mM mannitol both cause about a 2-fold reduction in the transcript level of eIF4a within about 6 h, according to the Arabidopsis gene expression atlas (eFP browser); so, for example, the apparent reduction in FBN11 mRNA levels in response to NaCl might be an artifact of altered eIF4a expression. Moreover, the sort of wild results with mannitol, showing induction and reduction, might have to do with the overlap of two kinetics: the rate of response of FBN11 transcription to mannitol versus that of eIF4a. Therefore, I urge the authors to consider using reference genes that are established to be insensitive to the conditions described here; several papers have outlined good RT-qPCR reference genes for abiotic stress and development. Moreover, I recommend using two unrelated reference genes in the future, so that results can be confirmed against two genes instead of relying on only one, potentially error-inducing, reporter.
The drought-response assays described in lines ~251-257 are very unusual; I’ve never seen this kind of experiment in the literature before, and I don’t think that severing shoots from roots and then putting them in a dry environment accurately reproduces biologically-relevant conditions. If I’m wrong, then this assay needs to be explained in better and more convincing detail, along with multiple citations from respected authorities in the field of drought stress tolerance supporting that this approach is useful.
I also noticed, in figure 6, that the results for the two alleles of fbn11 are very different (panel A, panel C in response to NaCl, etc.). Similarly, the two alleles aren’t really showing the same effects in Figure 7, although both are allegedly null alleles. fbn11-2 carries a T-DNA insertion in the gene’s 5’UTR, but this doesn’t always prevent transcription—actually, depending on the T-DNA, it might enhance expression of FBN11. I see the RT-PCR suggesting that full-length FBN11 is not made in this line, but then, why would the phenotypes not be identical? Complementation experiments would help to resolve this question, as would RT-qPCR to test for actual expression levels (it shouldn’t be zero).
Fig. 9 should consider the possibility that FBN11 is activated in response to other stress signals, such as salicylic acid (as previously reported—or show that this isn’t true!).
Author Response
Please see the attachment PDF for "response to reviewer2".
Thank you.

Round 2
Reviewer 1 Report
The authors have corrected most of the problems pointed in the previous report, improving the quality of the manuscript to a sufficient level.
Author Response
We are delighted to be able to meet the revision you requested.
Reviewer 2 Report
I am glad to see that the authors were able to quickly modify the text to address most of the concerns I raised in my last review.
My only major remaining concern is the use of eIF4a as a reference gene for RT-qPCR. The reference cited in the response to my review does not, in fact, use comparable conditions--that paper measured responses to phytohormones (GA and ABA). Indeed, eIF4a expression is not affected by GA and only slightly affected by ABA (<1.5-fold change in expression), whereas eIF4a expression is significantly affected by NaCl and mannitol at the concentrations used in this manuscript. The panel shown in the response to reviewers was not a quantitative PCR, nor even a semi-quantitative PCR (which would have minimal DNA to visualize a band), but appears to be an end-point PCR--which does not accurately reflect changes in mRNA levels.
Therefore, again, I strongly urge the researchers to repeat the experiment with an additional reference gene. If that is absolutely impossible, then I recommend at least noting in the text, throughout, including in figures, that expression is relative to eIF4a (so that readers know to interpret the results not as absolute values, but as relative effects).
Author Response
Response to reviewer2.
I am glad to see that the authors were able to quickly modify the text to address most of the concerns I raised in my last review.
(Addition) To explain why fbn11-1 has a weaker phenotype than fbn11-2 mutant in the mannitol and NaCl medium, we have identified the truncated FBN11 transcript generated by T-DNA insertion. Please see the newly added Figure S6. Along with this I have added lines at L444-446 as “This phenomenon suggests the possibility that truncated FBN11 produced by T-DNA insertion in fbn11-1 may play an incomplete role (Figure S6).”
My only major remaining concern is the use of eIF4a as a reference gene for RT-qPCR. The reference cited in the response to my review does not, in fact, use comparable conditions--that paper measured responses to phytohormones (GA and ABA). Indeed, eIF4a expression is not affected by GA and only slightly affected by ABA (<1.5-fold change in expression), whereas eIF4a expression is significantly affected by NaCl and mannitol at the concentrations used in this manuscript. The panel shown in the response to reviewers was not a quantitative PCR, nor even a semi-quantitative PCR (which would have minimal DNA to visualize a band), but appears to be an end-point PCR--which does not accurately reflect changes in mRNA levels.
Therefore, again, I strongly urge the researchers to repeat the experiment with an additional reference gene. If that is absolutely impossible, then I recommend at least noting in the text, throughout, including in figures, that expression is relative to eIF4a (so that readers know to interpret the results not as absolute values, but as relative effects).
(Answer) Thank you for your attentive review.
After comparing the results of Figure 5b with the results of the Arabidopsis gene expression atlas (eFP browser), it was confirmed that there was a difference in the expression pattern of FBN11 in 6 hr of NaCl. So, as you pointed out, we described in the results on page 10 at L333-338 as “When wild-type plants were treated with 150 mM NaCl, the expression of FBN11 was decreased at 6 hr (Figure 5B), but a contrast pattern appears in the Arabidopsis gene expression atlas (eFP browser) under the same conditions that showed an induced expression (Figure S3). The difference between these two experiments cannot rule out the possibility that the expression of the AteIF4a internal control gene was affected by NaCl treatment”.
We have also added few lines at L496-498 in the legend of figure 9 as “To determine the internal control for qRT-PCR analysis, the unaffected expression pattern of the AtelF4 gene was checked in wild-type and fbn11 mutant seed germination under mannitol treatment conditions (Figure S7)”.